# Evaluation of thrombophilia testing in the inpatient setting: A single institution retrospective review

Chun Ting Siu[1]*, Zachary Wolfe[2], Martin DelaTorre[1], Erafat Rehim[3], Robert Decker[2], Kathryn Zaffiri[1], Bradley Lash[2]

**1** Department of Medicine, Lehigh Valley Health Network, Allentown, Pennsylvania, United States of America, **2** Division of Hematology-Oncology, Department of Medicine, Lehigh Valley Health Network, Allentown, Pennsylvania, United States of America, **3** Department of Neurology, Lehigh Valley Health Network, Allentown, Pennsylvania, United States of America

\* chun.siu@lvhn.org

## Abstract

**Data Availability Statement:** All relevant data are within the manuscript and its Supporting information files.

### Background

Thrombophilia workup is typically inappropriate in the inpatient setting as testing may be skewed by anticoagulation, acute thrombosis, or acute illness.

### Objective

To determine adherence of inpatient thrombophilia testing with institutional guidelines.

### Patients and methods

A retrospective study to evaluate thrombophilia testing practices of adult patients who were admitted to Lehigh Valley Hospital at Cedar Crest with either venous thromboembolism or ischemic stroke in 2019. Testing included inherited and acquired thrombophilia. Patient charts were individually reviewed for three measured outcomes: 1) the number of appropriate thrombophilia testing in the inpatient setting; 2) the indications used for thrombophilia testing; 3) the proportion of positive thrombophilia tests with change in clinical management.

### Results

201 patients were included in our study. 26 patients (13%) were tested appropriately in accordance with institution guidelines and 175 (87%) patients were tested inappropriately. The most common reason for the inappropriate testing was testing during acute thrombosis. 28 of the 201 patients had positive thrombophilia tests, but the reviewers only noted 7 patients with change in clinical management—involving anticoagulation change.

### Conclusion

Our study revealed that a majority of inpatient thrombophilia testing did not follow institutional guidelines for appropriate testing and did not change patient management. These

**Funding:** The author(s) received no specific funding for this work.

**Competing interests:** The authors have declared that no competing interests exist.

thrombophilia tests are often overutilized and have minimal clinical utility in the inpatient setting.

## Introduction

The Virchow triad explains the pathogenesis of arterial and venous thrombotic disease in three broad categories: hypercoagulability, stasis, and endothelial damage. These categories can be due to acquired or genetic risk factors that predisposes patients into developing thrombosis. Inherited thrombophilia, or interchangeably called hereditary thrombophilia, are the genetic risk factors in which there are deficiencies of natural anticoagulants or genetic polymorphisms. Acquired thrombophilia are from environmental factors or a number of major medical illnesses such as cancer, myeloproliferative disorders, and antiphospholipid syndrome (APS). Testing can be done to determine if thrombophilia exist, but it is important to look at clinical risk factors when deciding on thrombophilia testing. This is highlighted by a statement from American Society of Hematology's Choosing Wisely Campaign, which recommends against thrombophilia testing in the setting of major risk factors such as surgery, trauma, or prolonged immobility [1]. Timing of thrombophilia tests also matters. Some thrombophilia tests can be affected by acute thrombosis and anticoagulation, which makes interpretation difficult in the inpatient setting.

Identifying inherited thrombophilia disorders rarely change management in patients with acute thrombosis. Current guidelines suggest that the initiation and intensity of anticoagulation for treating venous thromboembolism (VTE) should not be affected by thrombophilia testing [2]. The duration of anticoagulation is rarely affected by thrombophilia testing with the exceptions of severe thrombophilia and antiphospholipid syndrome. For instance, a meta-analysis examined the risk of recurrent VTE in two of the most common inherited thrombophilia—Factor V Leiden and Prothrombin Gene Mutation—finding only a modest increase in risk each heterozygous polymorphism as compared to patients without these genetic polymorphisms [3]. As such, the role of prolonged anticoagulation in these two types of inherited thrombophilia should be balance with the risk of bleeding. A special case that may warrant prolonged anticoagulation is antithrombin deficiency, due to a substantial increase in recurrent VTE [4]. However, testing in the inpatient setting has limited clinical utility because antithrombin testing may be affected by acute thrombosis and anticoagulation. Despite the limited value of inpatient thrombophilia testing, healthcare providers frequently order these tests which can add unnecessary cost to patient care.

We conducted a retrospective study to evaluate patterns of inpatient thrombophilia testing at our institution. The primary objective was to determine the appropriateness of inpatient thrombophilia testing that was ordered for patients admitted with acute thrombosis—either arterial or venous—based on our institutional guidelines. Secondary objectives were to examine the results of inpatient thrombophilia testing and the impact on clinical management, look at ordering practices by different hospital services, and determine the accumulated costs of inpatient thrombophilia testing.

## Materials and methods

### Study design and data collection

Our study was conducted at Lehigh Valley Hospital at Cedar Crest, which is a 729-bed tertiary care hospital with 48,296 inpatient and observation admissions during the 2019 fiscal year.

This hospital uses Epic Electronic Health Record (EHR). With EHR, we used ICD-10 diagnosis codes to retrospectively identify all patients 18 years or older who were admitted with venous thromboembolism (VTE) or arterial thrombosis over a one-year period in 2019. Patients who had inpatient thrombophilia testing were eligible for inclusion into this study. Patients were excluded if thrombophilia testing was ordered and/or performed as an outpatient.

After patients were identified for inclusion in the study, the EHR was queried to obtain patient demographic information, type of thrombosis, thrombophilia testing, and whether hematology-oncology service was consulted. Thrombophilia tests included were Antithrombin Antigen (ATAG), Antithrombin Activity (AT), Protein C (PC), functional Protein S (FPS), Protein S total antigen (PSTG), Factor V Leiden (FVLM), Prothrombin Gene Mutation (PTGM), Beta-2-Glycoprotein-1 Autoantibodies (B2G), Cardiolipin Autoantibodies (CL), Anticoagulant Sensitive PTT (aPTT), and dilute Russell's Viper Venom Time (dRVVT). These thrombophilia tests can be pulled from the query by the name of these tests and the CPT (Current Procedural Terminology) codes. In addition, our institution developed test panels to group thrombophilia disorders into 3 categories: inherited thrombotic risk panel (consist of AT, PC, FPS, FVLM, and PTGM), lupus thrombotic risk panel (LA, and dRVVT), and acquired thrombotic risk panel (B2G, CL, aPTT, and dRVVT). These test panels were identified by name and included in the query as well.

For chart review, two internal medicine resident physicians served as primary reviewer. First, the reviewers ensure each patient met the inclusion criteria. Second, the reviewers determine if testing adhered to institutional guidelines. Third, results of the thrombophilia tests were collected and reviewed to determine if there was a change in clinical management. A hematology oncology fellow guided the primary reviewers through a sample of patient charts to determine clinical relevancy in thrombophilia testing. Data was recorded using Microsoft Access.

This retrospective study was approved by the Institutional Review Board at Lehigh Valley Health Network. The IRB waived the requirement for informed consent. Reviewers did have access to patient identifying information to review the EHR chart.

## Definition of diagnoses

ICD-10 codes were used to identify the following diagnoses for inclusion in this study: deep vein thrombosis, pulmonary embolism, superficial vein thrombosis, splanchnic thrombosis, cerebral vein thrombosis, ischemic strokes, upper extremity and lower extremity arterial thrombosis. Arterial thrombosis due to myocardial infarction were excluded from the study.

## Definition of inappropriate testing

For this study, we used our institutional guidelines to determine the appropriateness of the testing that was ordered. These guidelines were previously developed through collaboration with hematology, neurology, internal medicine, and laboratory medicine and based off recommendations from major medical societies in the United States and United Kingdom [1, 2, 5, 6]. Routine thrombophilia testing is inappropriate for provoked VTE during acute thrombosis (defined as within 30 days), while on anticoagulation, within 2 weeks after discontinuation of anticoagulation, or in the setting of certain disorders such as malignancy, myeloproliferative disorders, or inflammatory bowel disorders. Provoking risk factors for VTE include surgery, trauma, prolonged immobility, pregnancy, puerperium, and hormonal therapy such as oral contraceptives.

## Definition of appropriate testing

Routine thrombophilia testing is appropriate under four categories: 1. Unprovoked VTE, Age <45, and ≥1 first-degree relative with VTE, 2. Recurrent VTE, 3. VTE in cerebral vein or splanchnic vein (portal, hepatic, or mesenteric), 4. Arterial thrombosis at Age <50. These four categories were formulated by our institution after evaluation of society guidelines and literature review [2, 5, 6].

## Determination of healthcare costs

Cost of thrombophilia tests was estimated from the 2019 Centers for Medicare and Medicaid Services Clinical Laboratory Fee Schedule and matched with the CPT codes used by our institution's laboratory.

## Data analysis

Data was exported from Microsoft Access and analyzed using Microsoft Excel for descriptive statistics. Using SPSS, Cohen's kappa coefficient was calculated in a sample of patients to quantify the inter-rater reliability between the two primary reviewers.

## Results

### Patient characteristics

During the one-year study period, Lehigh Valley Hospital at Cedar Crest admitted 3977 patients with either venous thromboembolism and/or ischemic stroke. 353 patients were identified from the EHR query to have inpatient thrombophilia testing. After manual review, 206 (5.1%) patients met our inclusion criteria. 5 patients were excluded from analysis because inpatient thrombophilia tests were ordered but discontinued prior to lab collection. The remaining 201 patients became the study population. Patient characteristics are summarized in Table 1. The median age was 55 (range 18–90) with near equal ratio of males (49%) and females (51%). Of the 201 patients, the ethnicity is predominantly White (79%). The three most common past medical history are >1 atherosclerotic risk factor (54%), prior stroke (14%), and cancer (11%).

### Inpatient thrombophilia testing

Table 2 illustrates the indications for thrombophilia testing. Only 26 patients (13%) of the 201 patients were tested appropriately as determined by our institutional guidelines. Fifteen patients met the indication for testing based on arterial thrombosis for patients age <50, 6 patient met the indication for recurrent VTE, 3 patients met the indication for unprovoked VTE at age <45 with family history of VTE, and 2 patients met the indication for unusual VTE sites. Of the 26 patients who were appropriately tested, 13 had positive results of which 7 were considered likely false-positive and the remaining 6 were considered positive. Only 4 out of those 6 patients had tests that changed patient management. Of these 4 patients, one patient was admitted for recurrent VTE, one patient with family history of systemic lupus erythematous (SLE) was admitted for unprovoked VTE, one patient with medical history of SLE was admitted for ischemic stroke, and one patient under age 50 was admitted for ischemic stroke. All four of these patients were found to have positive antiphospholipid antibodies and were appropriately switched to warfarin as the anticoagulation of choice. Hematology-Oncology service was consulted for 12 of the 26 appropriately tested patients and was responsible for ordering thrombophilia tests for 7 patients.

**Table 1. Patient characteristics.**

| | Patients (N = 201) | (%) |
|---|---|---|
| **Age, median** | 55 (18–90) | |
| **Gender** | | |
| Male | 99 | 49% |
| Female | 102 | 51% |
| **Race** | | |
| White or Caucasian | 160 | 80% |
| Black or African American | 20 | 10% |
| Multi-racial | 4 | 2% |
| American Indian or Alaska Native | 2 | 1% |
| Asian | 1 | 0% |
| Other | 8 | 4% |
| Unknown | 6 | 3% |
| **Past Medical History** | | |
| Atherosclerotic risk factor (eg, Hypertension, Hyperlipidemia, Smoking) | 109 | 54% |
| Atrial Fibrillation | 10 | 5% |
| Cancer | 22 | 11% |
| Autoimmune disease | 19 | 9% |
| Prior Deep Venous Thrombosis | 17 | 8% |
| Prior Pulmonary Embolism | 9 | 4% |
| Prior Stroke | 28 | 14% |
| **Length of Inpatient Stay in Days, mean** | 7 | |
| **Site of thrombosis at time of hospitalization** | | |
| Arterial Thrombosis other than Ischemic Stroke | 8 | 4% |
| Ischemic Stroke | 122 | 61% |
| DVT | 20 | 10% |
| PE | 40 | 20% |
| Multiple Thrombosis Sites | 6 | 3% |
| Portal Vein Thrombosis | 5 | 2% |

Inappropriate thrombophilia testing was seen in 175 patients (87%). Many patients meet multiple indications for inappropriate testing. Three of the most common inappropriate indications were acute thrombosis event (146 cases), arterial thrombosis at age >50 (82 cases), and patients currently on anticoagulation or within 2 weeks after discontinuation of anticoagulation (54 cases). Of the 175 patients tested inappropriately, 61 had positive results of which 39 were considered likely false-positive and the remaining 22 were considered positive. Only 3 out of those 22 patients had tests changed patient management. Of these 3 patients, one patient with history of malignancy was admitted for VTE and ischemic stroke, one patient was admitted for recurrent ischemic strokes, and one patient with history of VTE on Eliquis was admitted for ischemic stroke. All three of these patients were found to have antiphospholipid antibody positivity, so their anticoagulation of choice was affected. The patient with history of VTE on Eliquis admitted for ischemic stroke also had heterozygous prothrombin gene mutation, but that was not the reason for change in management. For the patient admitted with recurrent ischemic strokes, repeat testing in the outpatient setting revealed normal levels of cardiolipin antibodies, suggesting transient antiphospholipid antibodies. Hematology-Oncology service was consulted for 61 of the 175 inappropriately tested patients, and was responsible for ordering thrombophilia tests for 14 patients. Despite consulted by Hematology-Oncology

**Table 2. Thrombophilia testing indications and characteristics.**

| | Appropriate Indications | Inappropriate Indications |
|---|---|---|
| **Total (n = 201)** | 26 (13%) | 175 (87%) |
| **Appropriate Indications** | | |
| Arterial Thrombosis (Age<50) | 15 | - |
| Recurrent VTE | 6 | - |
| Unprovoked VTE (Age<45 and >/ = 1st relative VTE) | 3 | - |
| Unusual VTE sites[1] | 2 | - |
| **Inappropriate Indications[2]** | | |
| Acute Thrombosis Event (<30 days) | - | 146 |
| Arterial Thrombosis (Age>50) | - | 82 |
| Malignancy | - | 15 |
| On anticoagulation (currently or <2 wks after d/c) | - | 54 |
| Provoked VTE[3] | - | 11 |
| **Hematology Oncology Consult** | 12 (46%) | 61 (35%) |
| **Thrombophilia Test Ordered by Hospital Service** | | |
| Emergency Medicine | 1 (4%) | 6 (3%) |
| Family Medicine | 1 (4%) | 2 (1%) |
| Hematology Oncology | 7 (27%) | 14 (8%) |
| Medicine or Other Medicine Subspecialty | 4 (15%) | 48 (27%) |
| Neurology | 10 (38%) | 100 (57%) |
| OBGYN | 0 | 1 (1%) |
| Surgery or Surgery Subspecialty | 3 (12%) | 4 (2%) |
| **Number of Patients that Test Positive** | 6 (23%) | 22 (13%) |
| Change in type of anticoagulation | 4 | 3 |
| **Number of Patients that Test False-Positive** | 7 (27%) | 39 (22%) |
| Change in Management | 0 | 0 |

[1] Unusual VTE sites include: splanchnic veins (portal, hepatic or mesenteric), cerebral veins.

[2] A total of 175 patients had Inappropriate Indications. However, these indications are not mutually exclusive as some patients meet multiple criteria.

[3] Provoking risk factors: surgery, trauma, prolonged immobility, pregnancy, puerperium, hormonal therapy.

service on 61 patients, thrombophilia testing was still ordered inappropriately for 47 patients by other hospital services. Neurology service ordered more inappropriate tests (for 100 patients) than any other service combined. Those tests were ordered for patients being hospitalized for Ischemic Stroke—either acute, subacute, or chronic. Another group to frequently order inappropriate tests was Medicine or Other Medicine subspecialty service (for 48 patients).

The Cohen's kappa coefficient to measure inter-rater reliability on appropriate thrombophilia testing was 0.779 between the two primary reviewers (P<0.001).

## Healthcare cost with testing

From a cohort of 201 patients, there were 1049 inpatient thrombophilia tests performed. Of those tests, 922 were inappropriate tests with over 89% ordered as part of test panels. Test panels were developed by our institution with the intention of comprehensive testing. Table 3 illustrates the number of inappropriate thrombophilia tests ordered and their respective cost.

**Table 3. Inappropriate thrombophilia tests ordered and cost.**

| Name of Test | Affected by Anticoagulation and/or Acute Thrombosis | Estimated Cost Per Test[1] | Number of Test Ordered | Total Cost |
|---|---|---|---|---|
| Antithrombin Antigen | Yes | $12.01 | 1 | $12.01 |
| Antithrombin Activity | Yes | $13.17 | 6 | $79.02 |
| Beta-2 Glycoprotein Autoantibodies | No | $84.84 | 14 | $1,187.76 |
| Cardiolipin Autoantibodies | No | $84.84 | 17 | $1,442.28 |
| Factor V Leiden Mutation | No | $73.37 | 20 | $1,467.40 |
| Lupus Anticoagulant (dVRRT) | Yes | $10.64 | 13 | $138.32 |
| Protein C, Functional | Yes | $15.37 | 6 | $92.22 |
| Protein S, Functional | Yes | $17.03 | 6 | $102.18 |
| Protein S, Total Ag | Yes | $12.90 | 2 | $25.80 |
| Prothrombin Gene Mutation | No | $65.69 | 13 | $853.97 |
| **Test Panels** | | | | |
| Thrombotic Risk, Acquired Antiphospholipid (*4 individual tests*) | Yes | $130.43 | 101 | $13,173.43 |
| Thrombotic Risk, Acquired: Lupus (*2 individual tests*) | Yes | $17.31 | 30 | $519.30 |
| Thrombotic Risk, Inherited (*5 individual tests*) | Yes | $184.63 | 72 | $13,293.36 |
| **Total**[2] | | | 301 | $32,387.05 |

[1] Estimated Cost per Tests were calculated by matching the CPT codes from the 2019 Centers for Medicare and Medicaid Services (CMS) Clinical Laboratory Fee Schedule to the CPT codes used by our institution lab.

[2] A total of 175 patients had inappropriate thrombophilia testing. However, some patients had more than one test ordered.

Based on 2019 Centers for Medicare and Medicaid Services Clinical Laboratory Fee Schedule, the total estimated cost of the inappropriate thrombophilia tests were $32,387.05 dollars.

## Discussion

The objective of this retrospective study was to assess the adherence of inpatient thrombophilia testing to our institutional guidelines. We found a high rate of inappropriate testing. Over a 1-year period, only 13% of patients for whom testing was done had appropriate indications.

Based on published retrospective studies, inappropriate thrombophilia testing is common occurrence at other institutions with percentages ranging from 42 to 77% in their respective institutions [7–10]. Direct comparisons to other institutions can be misleading; however, inappropriate testing was higher at our institution [7–10]. The results in our study must be interpreted in context to the test panels created by our institution, which comprised most of the inappropriate testing. Notably, thrombophilia testing with test panels is what makes our study unique from other published retrospective studies. The test panels were created with the idea of ensuring comprehensive testing and as a convenient tool for healthcare providers. Instead, what we saw was the overuse of these test panels by other hospital services, despite recommendations by the hematology-oncology consult service.

This study also demonstrated that most inpatient thrombophilia testing did not change clinical management. Of the 201 patients tested, 28 patients had positive thrombophilia tests and only 7 patients had a change in management based on the test results, 21 patients were found to have inherited thrombophilia. Positive results for inherited thrombophilia testing in the inpatient setting was not shown to influence acute management in our study. This finding was seen consistently seen in other retrospective studies [7–10]. The remaining 7 patients had antiphospholipid antibody positivity which required a change in the type of anticoagulation to

reduce the risk of recurrent thrombosis. None of the testing influenced the duration of anticoagulation.

Thrombophilia testing is costly to patients and puts unnecessary burden on hospital resources. In 2019, there was a potential cost-saving of $32,387.05 dollars at our institution if inpatient thrombophilia testing had been appropriately ordered. Aside from cost, thrombophilia testing can also be potentially harmful to the patient. Patients can develop hospital-acquired anemia from excessive blood draws. Incorrect interpretation of test results may lead to inappropriate decisions regarding the type or duration of anticoagulation. We acknowledge several limitations of our single-center retrospective review. As previously discussed, our findings may not be applicable to all centers but the findings are consistent with previously published research from other institutions. Second, our institutional guidelines may differ from that of other institutions. Additionally, our ability during chart review to determine whether an event was provoked or unprovoked was often limited by the documentation of the treating provider. We also acknowledge that while this study is not the largest in size or longest in duration, the strengths of this study lie with its comprehensive overview. This analysis includes the type of thrombosis, the specific indications, the hospital services who ordered these tests, and a breakdown of the lab cost.

Multiple strategies are being considered at Lehigh Valley Hospital to encourage high-value care and minimize inpatient thrombophilia testing. For example, as seen in the University of Texas Southwestern Medical Center, a feedback system notifying ordering providers of inappropriate inpatient thrombophilia testing may discourage future inappropriate testing [7]. Another possibility, as seen at the University of British Columbia, is instituting a hard stop requiring printed requests for thrombophilia testing to reduce inpatient orders [11]. Other strategies include educational sessions on indications for thrombophilia testing and to discourage testing before hematology-oncology consultation. Recently, we incorporated a weblink to our institutional guidelines into the lab panels. Inpatient panel testing may need to be eliminated in order to encourage selective testing based on the clinical scenario. Finally, our institutional guidelines may need revision considering the low rate of testing that changed clinical management.

## Conclusion

This study demonstrate that thrombophilia testing is usually inappropriately ordered in the inpatient setting at Lehigh Valley Hospital at Cedar Crest. Thrombophilia test panels were frequently overused in the inpatient setting at our institution, which may have contributed to the high proportion of inappropriate testing. Inpatient thrombophilia testing rarely impacts acute management and the only test that altered management was antiphospholipid antibody testing.

## Supporting information

**S1 File. Data and calculations.** This contains the original data and calculations supporting the figures and tables displayed in the manuscript.
(XLSX)

## Author Contributions

**Conceptualization:** Chun Ting Siu, Zachary Wolfe, Erafat Rehim, Bradley Lash.

**Data curation:** Chun Ting Siu, Martin DelaTorre, Kathryn Zaffiri.

**Formal analysis:** Chun Ting Siu, Zachary Wolfe, Martin DelaTorre, Kathryn Zaffiri.

**Investigation:** Chun Ting Siu, Zachary Wolfe, Bradley Lash.

**Methodology:** Chun Ting Siu, Zachary Wolfe, Bradley Lash.

**Project administration:** Bradley Lash.

**Resources:** Kathryn Zaffiri, Bradley Lash.

**Software:** Chun Ting Siu, Kathryn Zaffiri.

**Supervision:** Zachary Wolfe, Bradley Lash.

**Validation:** Chun Ting Siu, Zachary Wolfe, Kathryn Zaffiri.

**Visualization:** Chun Ting Siu, Erafat Rehim, Bradley Lash.

**Writing – original draft:** Chun Ting Siu, Zachary Wolfe, Martin DelaTorre.

**Writing – review & editing:** Chun Ting Siu, Zachary Wolfe, Martin DelaTorre, Erafat Rehim, Robert Decker, Bradley Lash.

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
