## [Decision Letter · Decision Letter 0]

22 Jul 2021

PONE-D-21-14528

Evaluation of Thrombophilia Testing in the Inpatient Setting: A Single Institution Retrospective Review

PLOS ONE

Dear Dr. Siu,

Thank you for submitting your manuscript to PLOS ONE. After careful consideration, we feel that it has merit but does not fully meet PLOS ONE’s publication criteria as it currently stands. Therefore, we invite you to submit a revised version of the manuscript that addresses the points raised during the review process.

We look forward to receiving your revised manuscript.

Kind regards,

Michael Nagler, M.D., Ph.D., MSc

Academic Editor

PLOS ONE

Journal Requirements:

Reviewers' comments:

Reviewer's Responses to Questions

**Comments to the Author**

1. Is the manuscript technically sound, and do the data support the conclusions?

Reviewer #1: Partly

Reviewer #2: Partly

2. Has the statistical analysis been performed appropriately and rigorously? 

Reviewer #1: Yes

Reviewer #2: Yes

3. Have the authors made all data underlying the findings in their manuscript fully available?

Reviewer #1: Yes

Reviewer #2: Yes

4. Is the manuscript presented in an intelligible fashion and written in standard English?

Reviewer #1: Yes

Reviewer #2: No

5. Review Comments to the Author

Reviewer #1: Thank you for your manuscript entitled "Evaluation of Thrombophilia Testing in the Inpatient Setting: A Single Institution Retrospective Review". The topic of your manuscript is important given that thrombophilia testing is often overused and overvalued. Your manuscript is technically sound and most conclusions substantially correct.

However, I have some comments and suggestions:

-page 3: Your write that "duration of anticoagulation may not always be affected by results of thrombophilia testing". You should rather state that duration is barely affected by thrombophilia testing with the exceptions of severe thrombophilia and antiphospholipid syndrome

-p5 Use only one consistent abbrevation for anticoagulant sensitive aPTT (you use both LUA and LA). I would suggest to just use "aPTT" becouase "LA" stands for "Lupus anticoagulant" that can be detected with both aPTT and dRVVT.

-p7 Antiphospholipid antibodies are often only transiently positive. You should not use the term "false-postive" in this context.

-p8 use antithrombin activity instead of antithrombin III

-p5 and 8: "dilute Russell's viper venom": the test is in fact named "dilute Russell's Viper Venom TIME (dRVVT)).

-Given your data, you should at least discuss wether testing for hereditary thrombophilia (not APS!) in an inpatient setting is necessary at all. The most efficient way to reduce costs would be to just avoid inpatient testing for hereditary thrombophilia.

-Are your institution guidelines for thrombophlia testing valid for all hospital services or do some specific services (eg neurology) use their own guidance?

Reviewer #2: In a single-center retrospective study, the authors explored thrombophilia testing in inpatients and determined whether it was according to guidelines or not. This is an interesting study because little is known on this topic. However, the presentation can be significantly improved.

1.) Abstract:

The background can be sharpened and maybe even stated in one sentence.

Please state the aim of the study.

Please report details (Setting, inclusion and exclusion criteria more explicitely).

Definition of "Thrombophilia testing"

How where the data collected?

What was the criterion for "inappropriate" testing?

Testing "did not change management". How was this measured?

2.) Introduction:

Two issues are not discussed adequately: (1) What is the problem? You are correct that inpatient thrombophilia testing is often inadequate but is this sufficient to do the study? (2) What are aims and objectives of the study (this must match the problem stated before)? Please be precise and specific.

3.) Methods:

The methods are described very shortly and not very clearly arranged. I suggest including sub-headings (Design, setting and population/ definition of diagnoses / definition of inapropriate testing / data collection / determination of healthcare costs/ statistical analysis). Please follow the STROBE reporting guidelines.

Regarding data collection: how was the chart review done? How where the patients screened?

Database query for patient screening: is it valid and complete?

4.) Results:

I suggest sub-heading to improve clarity (patient characteristics...)

How many patients were admitted with VTE or arterial thrombosis (overall) and what was the proportion of patients with thrombophilia testing. This would be easy to determine and would strengthen the manuscript.

5.) The discussion would benefit from structuring. An established structure is: (1) What did we find? (2) discussion of the results in context of previous literature (3) strengths and limitations (4) what does it mean (5) conclusions.

6.) Conclusions:

"In summary, this retrospective study showed that inappropriate thrombophilia testing is common in the inpatient setting." Please modify this first sentence to make it clear that it only refers to your institution.

"Based on published retrospective studies, this issue is also seen in other institutions as well." This sentence fits into the discussion, not the conclusion. Please re-word.

"Inpatient thrombophilia testing rarely affects acute management and should be recommended only in selected patients." Was this observed in your study??

"The ordering practices revealed the overuse of thrombophilia lab panels.." Was this observed in your study??

6. PLOS authors have the option to publish the peer review history of their article (what does this mean?). If published, this will include your full peer review and any attached files.

Reviewer #1: No

Reviewer #2: **Yes: **Michael Nagler

---

## [Author Response · Author response to Decision Letter 0]

24 Aug 2021

Please see cover letter containing our response to the reviewers.

---

## [Editor Report · Decision Letter 1]

8 Sep 2021

Evaluation of Thrombophilia Testing in the Inpatient Setting: A Single Institution Retrospective Review

PONE-D-21-14528R1

Dear Dr. Siu,

We’re pleased to inform you that your manuscript has been judged scientifically suitable for publication and will be formally accepted for publication once it meets all outstanding technical requirements.

Kind regards,

Michael Nagler, M.D., Ph.D., MSc

Academic Editor

PLOS ONE

Additional Editor Comments (optional):

All comments have been adequately addressed.
---

## [Editor Report · Acceptance letter]

10 Sep 2021

PONE-D-21-14528R1 

Evaluation of thrombophilia testing in the inpatient setting: a single institution retrospective review 

Dear Dr. Siu:

I'm pleased to inform you that your manuscript has been deemed suitable for publication in PLOS ONE. Congratulations! Your manuscript is now with our production department. 

Kind regards, 

on behalf of

Prof. Dr. Michael Nagler 

Academic Editor

PLOS ONE